# Study for Evaluation of Hydrogels after the Incorporation of Liposomes Embedded with Caffeic Acid

**DOI:** 10.3390/ph15020175

**Published:** 2022-01-31

**Authors:** Ioana Lavinia Dejeu, Laura Grațiela Vicaș, Lavinia Lia Vlaia, Tunde Jurca, Mariana Eugenia Mureșan, Annamaria Pallag, Georgeta Hermina Coneac, Ioana Viorica Olariu, Ana Maria Muț, Anca Salomea Bodea, George Emanuiel Dejeu, Octavian Adrian Maghiar, Eleonora Marian

**Affiliations:** 1Doctoral School of Biomedical Science, University of Oradea, 1 University Street, 410087 Oradea, Romania; ioana.dejeu@gmail.com (I.L.D.); salomea.bodea@gmail.com (A.S.B.); 2Faculty of Medicine and Pharmacy, University of Oradea, 1 Decembrie Street, 410073 Oradea, Romania; jurcatunde@yahoo.com (T.J.); marianamur2002@yahoo.com (M.E.M.); annamariapallag@gmail.com (A.P.); dejeu.george@gmail.com (G.E.D.); octimaghiar@gmail.com (O.A.M.); marian_eleonora@yahoo.com (E.M.); 3Department II—Pharmaceutical Technology, Formulation and Technology of Drugs Research Center, “Victor Babeș” University of Medicine and Pharmacy, Eftimie Murgu Sq. no. 2, 300041 Timișoara, Romania; coneac.georgeta@umft.ro (G.H.C.); mut.anamaria@umft.ro (A.M.M.); olariu.ioana@umft.ro (I.V.O.)

**Keywords:** caffeic acid, liposomes, hydrogels, carbopol, rheological properties

## Abstract

Caffeic acid (CA), a phenolic acid, is a powerful antioxidant with proven effectiveness. CA instability gives it limited use, so encapsulation in polymeric nanomaterials has been used to solve the problem but also to obtain topical hydrogel formulas. Two different formulas of caffeic acid liposomes were incorporated into three different formulas of carbopol-based hydrogels. A Franz diffusion cell system was used to evaluate the release of CA from hydrogels. For the viscoelastic measurements of the hydrogels, the equilibrium flow test was used. The dynamic tests were examined at rest by three oscillating tests: the amplitude test, the frequency test and the flow and recovery test. These carbopol gels have a high elasticity at flow stress even at very low polymer concentrations. In the analysis of the texture, the increase of the polymer concentration from 0.5% to 1% determined a linear increase of the values of the textural parameters for hydrogels. The textural properties of 1% carbopol-based hydrogels were slightly affected by the addition of liposomal vesicle dispersion and the firmness and shear work increased with increasing carbomer concentration.

## 1. Introduction

Caffeic acid (CA) is a hydroxycinnamic acid, which belongs to the class of phenolic acids, is present in various plant products (coffee beans, apples, potatoes, cabbage, olive oil, wine, tea, berries) [1,2], having important antioxidant, anti-inflammatory, antimicrobial, antiviral [3], antithrombotic [4], and anticancerous properties [5]. It can be used as an ingredient in the cosmetic industry, in drugs, or in dietary supplements [6]. Some studies have shown the potential of CA in the treatment of inflammatory skin pathologies, such as psoriasis. Moreover, due to its antioxidant power, CA has photoprotective action against DNA damage, protecting the skin against aging and preventing melanoma [7,8,9,10].

Due to its low solubility in water, caffeic acid has limited use, so encapsulation in polymeric nanomaterials has been used to solve this problem. In addition, the encapsulation of caffeic acid helps to improve bioavailability, absorption, and mechanism of action [11,12]. 

Phenolic compounds have an excellent penetration into the skin which makes them good candidates for effective treatment in dermatological diseases [13]. The encapsulation of phenolic compounds in liposomes leads to an improvement of their solubility, in vitro stability, and in vivo bioavailability [14,15,16,17] and also ensures protection of the molecules against oxidation because the liposomes form a barrier around the content, as well as a resistance against the action of enzymes [18].

From the group of particulate systems, liposomes are closed structures, consisting of one or more lipid layers organized between two hydrophilic compartments. These phospholipids underlying the structure of liposomes are amphiphilic and consist of a polar (hydrophilic) and a nonpolar (hydrophobic) part, which leads to the possibility of forming bonds with both hydrophilic and hydrophobic agents, thus representing a very big advantage in the role of being a transporter molecule. It was observed that the encapsulation of antitumor substances in different types of liposomes led to the improvement of the specificity, bioavailability, and biocompatibility properties of these substances. In the last two decades, significant efforts have been made to exploit liposomes for therapeutic purposes [19].

Direct release to the site of action of substances after absorption in the epidermis allows the achievement of local tissue concentrations and thus the incidence of severe systemic side effects is reduced compared to oral administration because the drug reaches the systemic circulation in a very small amount [20]. In order to avoid the appearance of these side effects, but also to obtain a faster healing in various dermatological diseases, nanotechnology is being increasingly used [21].

Hydrogels are three-dimensional cross-linked networks that can retain, expand, and transport large volumes of water, due to their hydrophilic nature [22], thus being suitable for biological applications. The gel structure is formed in contact with water and can be obtained from natural, semi or synthetic polymeric materials [23]. They are chemically stable, biodegradable, protect the labile drug from degradation, and can control the release of small molecule drugs, or macromolecules [24]. Carbopol is a cross-linked acrylic polymer used in hydrogel preparation in the cosmetic and pharmaceutical industries [25]. It is a transparent, stable, and non-toxic polymer with thickening properties, non-irritating, being suitable for gel preparations [26].

## 2. Results

### 2.1. Preparation of Hydrogels

CA, a phenolic acid, in contact with various environmental factors, can undergo oxidation reactions, changing its color and causing a decrease in therapeutic activity. Therefore, its use in topical and oral administration becomes inefficient [14,15]. CA also has low solubility in cold water (less than 1 mg/mL at 22 °C) so its bioavailability is low and absorption is difficult. The chemical compartmental structure of liposomes is similar to that of normal human cell membranes. Liposomes can store, protect and transfer substantial amounts of drugs while being well tolerated by the body. Thus, the use of liposomes as a carrier system is based on the fact that the entire liposome content is protected against chemical degradation. The potential of this technology is great, considering that encapsulation, protection, and delivery of AC will increase its bioavailability.

The preparation formulas of the hydrogels are presented in Table 1.

To achieve topical preparations with efficient administration, 12 hydrogel formulations were prepared whose components are biocompatible and similar to biological tissues. Of the 12 formulas, in 6 we incorporated liposomes loaded with caffeic acid and 6 formulas without liposomes were used as control samples. The complete analysis including the characterization of the macroscopic and rheological properties of hydrogels, their pH and the release of CA trapped in liposomes from these preparations will provide useful information for optimal application strategies on skin tissues.

### 2.2. Determination of Macroscopic Properties and pH

First, an evaluation of the macroscopic characteristics of the prepared formulations was performed immediately after preparation. Thus, all the prepared hydrogels had a homogeneous, translucent appearance, and yellowish-brown color (data are depicted in Appendix A, Appendix A). 

The pH was determined potentiometrically, using a pH meter type pH 7310 Inolab (Xylem Analytics GmbH, Wellheim, Germany). The pH of the prepared hydrogels was between 4.70 and 4.90. When preparing the gels, our goal was to prepare formulations with a pH close to the skin’s natural surface pH, which is around 4.7 [27]. Regarding gel preparation, proper pH adjustment is very important to initiate gel formation. The most favorable pH range for this is around 6.5–7.0 [28]. Polymers in the group of non-neutralized polycarboxylic acids usually have a pH range of 2.5–3.5 depending on the concentration of the polymer. These non-neutralized dispersions have very low viscosi ties, so the addition of triethanolamine is necessary to increase the pH [29]. The optimum viscosity can be reached in pH ranges of 5.5–7.0.

### 2.3. Viscoelastic Measurement of Liposomal Hydrogels

#### 2.3.1. The Steady-State Flow Test

From the rheograms and viscosity curves of the prepared hydrogels (shown in Appendix A, Appendix A), the increase of the shear stress may be noticed and, respectively, the decrease of the viscosity with the increase of the deformation speed, which reflects the fact that all the tested preparations are part of the category of non-Newtonian systems, with pseudoplastic behavior and yield stress. 

The rheological behavior of experimental hydrogels was modeled using four rheological models: Ostwald de Waele (Equation (1)), Herschel–Bulkley (Equation (3)), Casson and Bingham. With these models, the shear stress values obtained according to the shear speed on the ascending portion of the rheograms were fitted. As the accuracy of fitting the data with the Casson and Binghan models was poor (*r* < 0.9) for all tested formulations, the paper does not present the equations of the models and the values of the parameters obtained by fitting with these models. Table 2 shows the mean values of the parameters of the Ostwald de Waele and Herschel–Bulkley models, calculated by the non-linear regression method.

From Table 2, it can be seen that the Herschel–Bulkley model described better than the Ostwald de Waele model the flow behavior of the studied hydrogels, revealed by the higher values of the correlation coefficient *R*.

The values of the apparent viscosity measured at the shear speed of 100 1/s for all studied hydrogels are listed in Table 3. Based on the values of this parameter, the control hydrogels in both series (containing 0.5% and respectively 1% polymer) can be ranked as follows: FG I blank > FP I blank > FA I blank and FG II blank > FP II blank > FA II blank.

By incorporating drug liposomes into hydrogels based on 0.5% carbomer, there is a marked decrease in their apparent viscosity: 7.7 times in the case of formulations containing glycerol or isopropyl alcohol and 11.2 times in the case of formulations containing propylene glycol. In contrast, the viscosity of liposomal hydrogels containing 1% Carbopol 940 decreased only 1.4–1.9 times compared to the corresponding hydrogel bases, the decrease being more pronounced in the case of propylene glycol formulation.

Thixotropy is a property specific to pseudoplastic materials whose viscosity decreases under the action of shear stress as a result of structure destruction, followed by the progressive restoration of viscosity and restoration of the structure when shear stress stops [30,31]. This property is desirable for semi-solid preparations, including hydrogels, as it indicates that when applied to the skin (under the action of a shear stress), the semi-solid product becomes thinner, and easier to spread.

The rheograms and viscosity curves confirm that both hydrogel bases and liposomal hydrogels showed varying degrees of thixotropy, i.e., a reversible, time-dependent decrease in viscosity after the application of a constant shear rate. The values of the thixotropy index, calculated for the hydrogel bases with 0.5% Carbopol 940, were small (26.49–36.41%), decreasing in the following order: FG I blank > FP I blank > FA I blank (Table 4). The doubling of the polymer concentration in the hydrogel bases determined the considerable increase (2.58 times and 2.96 times, respectively) of their thixotropy index only in the case of formulations containing propylene glycol or isopropyl alcohol. In contrast, the thixotropy index of the blank FG II formulation was only 2 units lower than that of the blank FG I formulation (Table 3). Based on these results, it can be suggested that the effects of the cosolvent on the structure of the polymer network and implicitly on the gel thixotropy are manifested especially at low carbomer concentrations (0.5% in the present study). In the case of 1% carbomer systems, the polymer network is highly structured and less sensitive to the cosolvent in the composition.

Similar to viscosity, the thixotropy index of hydrogels by 0.5% Carbopol 940 decreased drastically by loading with liposomes, reaching values very close to the minimum limit (5%); the values of this parameter were about 6 times lower than those of the control hydrogels containing glycerol or propylene glycol, and in the case of the formulation FA I (with isopropyl alcohol) the difference compared to the control hydrogel was about 3.5 times smaller. In contrast, the thixotropic character of hydrogels based on 1% polymer and propylene glycol, isopropyl alcohol, or glycerol was affected in a lesser measure or not at all by the incorporation of liposomes.

The differences between the hydrogel bases in the two series in terms of consistency, viscosity, and thixotropy can be attributed to the different polarity of the cosolvents used in the formulation, which causes variations in the parameter’s degree of hydration and solubility parameter of the polymer by changing the attractive forces of the intermolecular polymer-solvent. In water, in the absence of the cosolvent and after neutralization with a base, the ionized carbomer chains are expanded as a result of electrostatic repulsion and form a rigid network in which water molecules will be retained by hydrogen bonds (attraction forces) between them and the carboxyl groups of the carbomer. The hydration of macromolecules is strong and the mobility of water is reduced, increasing the viscosity and consistency of the system. In a cosolvent-water mixture with a lower polarity, the solubility of the less ionized polymer chains decreases, and they will be more or less tightly packaged, preferentially interacting with each other, to the detriment of interaction with water molecules. They leave the crosslinked network of the polymer, being attracted in part by the molecules of the hydrophilic cosolvent (alcohol or polyol), with the formation of new hydrogen bonds. As a result, the degree of hydration of the macromolecules decreases, while decreasing the consistency and viscosity of the hydrogel. The lower the polarity of the cosolvent, the lower are the viscosity and consistency of the gel, through these mechanisms [32,33]. In the case of the studied hydrogel bases, the polarity of the aqueous liquid medium was decreased by the addition of a cosolvent with a dielectric constant *ε* lower than water (43 for glycerol, 32 for propylene glycol, and 17.9 for isopropyl alcohol). As a result, by the mechanisms described above, their viscosity/consistency decreased in the same order.

#### 2.3.2. Dynamic Tests (Oscillators)

Compared with steady-state rheological tests, oscillatory tests better correlate rheological properties with the microstructure of systems, as they can be examined at rest, without disturbing their underlying structures. The viscoelastic behavior of carbomer-based hydrogels, containing liposomes loaded with caffeic acid, was assessed by three oscillatory tests: the oscillation amplitude sweep test, the oscillation frequency sweep test, and the flow and recovery test.

The oscillation amplitude sweep test allows the identification of the linear viscoelastic region (LVR), which indicates the interval in which an oscillatory test can be performed without destroying the sample structure. In the linear viscoelastic region, the stress is directly proportional to the deformation, and the viscoelastic modules (accumulation modulus *G’*, and loss modulus *G’’*) remain constant. The results of this test are usually presented in the form of profiles of viscoelastic modules *G’* and *G’’* depending on the shear stress. In this diagram, the LVR region appears on the left, in the range with the lowest stress values. For evaluation, the curve of the elastic modulus (accumulation) *G’* is preferred, as it can provide information about the structural state of the polymer dispersion under at rest conditions. The accumulation module characterizes the degree of cohesion of the polymeric networks established following the entanglement or interpenetration of macromolecules by the effect of weak attractive forces (Van der Waals type) [34]. In the LVR region, this module has a constant value, the so-called plateau value. The linearity limit (limit value of the LVR region) is determined by the rheometer software and indicated in the diagram by a vertical line.

Figure 1, Figure 2 and Figure 3 show the results of the amplitude sweep to which the studied liposomal hydrogels were subjected, to identify the LVR region, and Table 4 lists the “plateau” values of the two dynamic modules.

It is observed that the response of all liposomal hydrogels was linear in the range of low shear stress values (0.5–10 Pa), in which the values of the two dynamic modules remained constant, being independent of the shear stress. In the LVR region, the imposed stress is lower than the strength of the bonds that support the microstructure of the systems and as a result, they deform elastically, but their integrity is not affected. In the field of linear viscoelasticity, the profiles of the two modules were parallel and without a point of intersection. Moreover, in this field, for all carbomer-based hydrogels analyzed (whether or not loaded with liposomes), the accumulation modulus was 10–15 times higher than the loss modulus (*G’* > *G’’*), a characteristic relation for elastic solids [35]. The difference of more than one order of magnitude between the values of the two modules is an indicator of their structural stability under the action of mechanical forces. Table 4 shows that all carbopol hydrogels not loaded with liposomes showed values of accumulation and loss modules that were quite close, the formulation variables having a low influence. Thus, in the 0.5% polymer series, the propylene glycol containing hydrogel resulted in a lower modulus value of *G’*, indicating a slightly lower degree of cohesion of the polymer networks than that of the glycerol or isopropyl alcohol hydrogels. However, by doubling the carbopol concentration, the *G’* modulus calculated for the control hydrogel with propylene glycol was increased, exceeding the values of the *G’* modules of the other two control formulations (with glycerol and isopropyl alcohol). In contrast, the addition of caffeic acid liposomes to hydrogels based on 0.5% Carbopol 940 resulted in a considerable decrease in the values of the accumulation modulus (5.85 times for glycerol hydrogel, 7.8 times for propylene glycol, and 5.25 times for with isopropyl alcohol), which reflects the decrease in the degree of cohesion of the polymer networks. On the other hand, the microstructure of 1% Carbopol 940 based hydrogels was much less affected by the addition of liposomes, which was revealed by the slight decrease (1.12–1.55 times) in the values of modulus *G’* (Table 4).

The frequency spectrum, obtained by performing the frequency sweep test, can be considered an “imprint” of the existing microstructure in the analyzed system [36]. The frequency sweep test allows the study of the viscoelastic character of a gel, reflected both by the rigidity (elasticity) determined by the presence of the three-dimensional network formed by the polymer chains, and by the flexibility (viscous behavior). In the frequency sweep test diagrams, the elastic modulus *G’* provides information on the stability and structural strength of the gel, and the viscous modulus *G’’* informs about its flexibility (viscous behavior). The higher the accumulation modulus, the higher is the rigidity of the viscoelastic gel (mechanical and structural stability is higher) [36]. The results of the frequency sweep are shown graphically in Figure 4, Figure 5 and Figure 6, in the form of the curves of the modulus elastic (*G’*) and viscous (*G’’*) as a function of the oscillating frequency.

For all studied hydrogels (unloaded and loaded with liposomes) it is found that both *G’* and *G’’* modules increased in frequency (the increase was linear in the range of 0.5–50 Hz), showing the viscoelastic character of the systems. Throughout the selected frequency range, the values of the elastic modulus (*G’*) were much higher than those of the viscous modulus (*G’’*), indicating the preponderance of the elastic character and the crosslinked structure of all the hydrogels analyzed. Several differences can be attributed to the formulation variables of the hydrogel base and the incorporation of liposomes in it. Thus, in the case of hydrogels not loaded with liposomes, the doubling of the polymer concentration caused a slight increase in the values of the elastic modulus (especially in the range of higher frequency values), while the values of the loss modulus did not change significantly (Figure 4a,c, Figure 5a,c and Figure 6a,c). This fact has been demonstrated by other studies [37].

The comparison of the curves of the elastic modulus *G’* produced by the analyzed hydrogels also illustrates the influence of the type of cosolvent in the formulation. In the case of each series of hydrogel bases (containing carbopol in the same concentration but different cosolvent), the curves of the *G’* module overlapped or were almost superposable over the whole frequency range studied, suggesting the low dependence of the viscoelasticity of the cosolvent polarity systems, present in a concentration of only 20% (the majority solvent is water). The liposomal hydrogels based on 0.5% carbomer behaved similarly in the frequency sweep test, but over a range of *G’* values of about 4–5 times smaller. In contrast, Figure 4d, Figure 5d, and Figure 6d illustrate that those based on 1% carbomer produced different and higher values of the *G’* modulus over the whole frequency range studied (the curves were not superposable). The positioning of the *G’* modulus curve corresponding to the glycerol-containing liposomal hydrogel (FG II formulation) can be observed above the curves obtained for the other two liposomal preparations containing isopropyl alcohol and propylene glycol (FA II and FP II formulations). This shows that the presence of glycerol slightly increases the *G’* modulus and causes the formation of a more elastic microstructure compared to isopropyl alcohol and propylene glycol, cosolvents with lower polarity. It should be noted that the values of modulus *G’* produced by the FA II formulation were slightly higher than those obtained for the FP II formulation (Figure 5d and Figure 6d), revealing that in the presence of isopropyl alcohol the elasticity of the liposomal hydrogel structure is higher than in the presence of propylene glycol, although it is more polar. These results suggest a more pronounced influence of the solvent composition of the hydrogel base on the viscoelasticity of the final preparation. The effect can be attributed to the modification of the crosslinked structure formed by the polymer chains with the change of the polarity of the vehicle, implicitly of its capacity to dissolve the polymer. It is known that the elastic modulus is closely correlated with the connectivity of the polymer network and is directly proportional to the chains of the elastic active network [37]. In general, the degree of crosslinking of the structure of neutralized carbopol gels and their viscoelasticity increases with increasing polymer concentration and solvent polarity, factors responsible for the presence of polymer chains in a more extensive form [38].

On the other hand, by incorporating the liposomes loaded with caffeic acid in the studied Carbopol 940 hydrogels, the values of the *G’* modulus decreased significantly, the effect being more pronounced in the case of systems containing 0.5% carbopol (Figure 4b,d, Figure 5b,d and Figure 6b,d). The decrease in the values of the accumulation modulus reveals a less elastic behavior of the liposomal hydrogels compared to that of the respective hydrogel bases. Unlike the elastic modulus, the values of the viscous modulus *G’’* of the hydrogel bases were less affected by the incorporation of liposomes with caffeic acid. In Figure 4b,d, Figure 5b,d and Figure 6b,d it can be seen that the respective profiles are very close and are positioned in the range of low values of the module *G’’* (5–200 Pa). The lower rigidity of liposomal hydrogels (compared to control hydrogels) can be explained by the fact that the liposomal particles loaded in the gel cannot deform as easily or more easily than the control gel under the action of the applied stress [39].

Figure 4, Figure 5, Figure 6 and Figure 7 show that the complex viscosity (η*) of all studied hydrogels decreased with increasing oscillation frequency, indicating their pseudoplastic behavior [40], which was also demonstrated by the steady-state flow test.

#### 2.3.3. Determination of Yield Stress

Figure 8, Figure 9 and Figure 10 illustrate the evolution of the viscoelastic modules *G’* and *G’’* under the action of shear stress, an evolution in which the yield stress can be highlighted, noting the value of the shear stress at the limit of the LVR region (linearity limit), for which the product begins to soften (to flow).

#### 2.3.4. Texture Analysis

Texture analysis is an instrumental method that provides information on a material’s response to an external force and is based on the conversion of force measurement (quantitative parameter) into qualitative organoleptic parameters, thus making an approximate sensory description of materials [41]. In the case of semi-solid preparations for application on the skin, texture analysis provides a reliable picture of their mechanical properties, such as ease of removal from the conditioning container, ability to spread on the skin, prolonged contact time at the site of administration (bioadhesion) and acceptable viscosity [42,43]. In addition, texture analysis is a particularly useful method to compare preparations of the same origin, tested under the same conditions. Thus, the method is a valuable support in the development stage of the formulation of semi-solid preparations, including hydrogels, because their composition determines their textural properties.

The back compression-extrusion test is sensitive and allows the determination of four important textural parameters (hardness or firmness, consistency, cohesiveness, and viscosity index) (shown in Appendix A, Appendix A), due to the large contact surface of the test probe with the sample to be analyzed.

The hardness or firmness of the sample is indicated by the maximum force (in g) measured when the probe enters the sample. The downward movement of the probe also allows the calculation of the compressibility of the sample (in g·s) indicated by the total mechanical work performed by the probe (force) to penetrate the sample to a certain depth. Firmness and compressibility quantify the deformation of the sample at compression and shear, being correlated with its consistency and density. The higher the firmness and compressibility values, the higher the consistency and density of the sample [44,45]. Low values of these parameters are correlated with the ease of removing the preparation from the container and applying it to the surface of the skin. During the upward movement of the test probe, two other textural parameters of the sample are determined: cohesiveness (in g) and viscosity index (in g·s). The cohesiveness is indicated by the maximum (absolute) negative force measured during the withdrawal of the test probe from the gel and is the expression of the gel’s ability to restore its structure after application. The more negative the force value, the more cohesive (viscous) the sample. A high value of cohesiveness increases the performance of the product at the application site, allowing the complete restoration of the gel structure after administration. The viscosity index, also known as adhesiveness, is determined as the area under the curve on the negative side of the graph, expressing the mechanical work required to remove the test probe from the gel. The viscosity index is the expression of the cohesive forces between the molecules in the gel. The higher the value of this parameter, the more energy is required to overcome the resistance of the sample to flow from the disk as the probe withdraws from the gel [46]. Topical hydrogels with a prolonged adhesion at the site of administration (favorable to the therapeutic result), will present higher values of the viscosity index [47,48].

#### 2.3.5. Determination of Consistency by the Penetrometric Method

The results of the penetrometric measurement of the consistency (more precisely of the hardness) of the studied hydrogels are presented in the form of histograms in Figure 11.

From Figure 11 it can be seen that the control hydrogel formulations FG II blank, FP II blank and FA II blank were the most compact, producing the lowest values of the degree of penetration (131.0 ± 2.94 mm, 129.3 ± 2.87 mm, and 133.0 ± 3.27 mm, respectively), as expected, due to the presence of Carbopol 940 gelling agent in a concentration of 1%. Insignificant differences between the three values of the degree of penetration indicate that the type of cosolvent in the composition of the hydrogel base did not affect its consistency. In contrast to these control formulations, the 0.5% Carbopol 940 liposomal hydrogels, namely FG I, FP I, and FA I, were more fluid, producing the highest penetration values. Compared to these, the control hydrogels with 0.5% Carbopol 940 were more consistent, showing values approximately 3 times higher for the degree of penetration. Consequently, it can be stated that the incorporation of liposomes loaded with caffeic acid in hydrogel bases containing 0.5% Carbopol 940 also affected their consistency, along with other rheological properties (viscosity, thixotropy, viscoelasticity, yield stress). In the case of liposomal hydrogels based on 1% carbomer, the penetrometric test showed, similar to the other rheological tests performed, the low influence of liposomal particles with caffeic acid on the consistency of the hydrogel base, indicated by the close values of the degree of penetration. The exception was liposomal hydrogel based on 1% carbomer and propylene glycol, whose consistency was about 2 times higher than that of the hydrogel base, most likely due to the presence of propylene glycol as a cosolvent.

#### 2.3.6. Spreadability Capacity Determination

The test results of the tensile capacity of experimental hydrogels based on Carbopol 940 are presented in Table 5, according to which dependence of the parameters measured by the formulation variables (polymer concentration, cosolvent type, and presence of liposomes with caffeic acid) is observed.

Firmness and shear work increased with increasing carbomer concentration, as a result of increasing adhesion forces inside gels (increasing the interaction between polymer molecules in the three-dimensional network).

### 2.4. The Release Studies

The in vitro release profiles of liposome formulations were investigated with the Franz diffusion method. Figure 12 shows the results of in vitro permeation profiles of CA through the membrane impregnated with the receptor solution. Six independently filled alcohol cells (30%) were used during the experiments. The permeation profiles of the active ingredients showed dependence on the basic type of gel used. 

Samples were taken from Franz cells at different times (0.5, 1, 2, 3, 4, 5, 6, 7, 8, 12 h) to determine the CA released from the synthesized hydrogels. The calibration curve for CA was used to interpret the results: *y* = 0.0644*x* + 0.0365, *R*^2^ = 0.9997, where *y*—absorbance of the solution (u.a) at 325 nm, and *x*—concentration in CA (mmol/L).

The synthetic porous membrane used for Franz cell testing allowed the passage of CA which was released from carbopol-based hydrogels. The cumulative amounts of CA that were released and penetrated the surface of the synthetic membranes were represented graphically according to time. The data are shown in Appendix A, Appendix A. 

It can be seen that free CA is released in a percentage of 92.32% in the first four hours, while from hydrogels containing carbopol in a concentration of 0.5% (FG I, FP I, FA I) the percentages in which CA is released are: 89.27%, 87.69%, 88.40% in the first 8 h, and from hydrogels containing a double amount of carbopol (FG II, FP II, FA II) the percentages in which CA is released are: 86.29%, 86.19%, 84.98% in the first 12 h.

## 3. Discussion

Carbopol, a crosslinked polyacrylic acid polymer extremely effective in modifying its rheological properties, capable of providing a high viscosity by forming clear gels, was used in the preparation of semi-solid pharmaceutical forms of the hydrogel type.

Based on the experimental results, it can be concluded that when testing the performance of hydrogels containing CA, the concentration of carbopol in the formula influences the release of the substance. Thus, the higher the amount of carbopol used in the formulation of the hydrogels, the greater the release of CA from the liposomes in the hydrogels over a longer period.

The alcohols used as cosolvents in the preparation of hydrogels did not significantly influence the release of CA from the obtained hydrogels. The presence of liposomes in the formulas did not significantly change the pH values for the hydrogels. 

The yield stress indicated the time-dependent change of the shear stress at constant deformation speed, the downward curve being located below the upward one, which reveals the slightly thixotropic character of the preparations. 

The values of the flow point (*τ0*) calculated for all formulations with the help of the Herschel–Bulkley model are negative, having no physical significance. As a result, the Herschel–Bulkley model is not convenient for the description of the analyzed systems [49]. The Ostwald de Waele rheological model are used for this purpose, which fits well with the experimental data, the values *R* > 0.9 (Table 3). The parameters of this model (*K* and *n*) can be correlated with the formulation variables (carbomer concentration and cosolvent nature). The flow index *n* obtained for all formulations was <0.5, the values falling in the range 0.173–0.261 and reflecting their pseudoplastic non-Newtonian character. In each series of hydrogels containing the same cosolvent, there were no minor variations in the concentration of the polymer. Thus, for the control formulations (not loaded with liposomes) the *n* values were identical (in the case of the glycerol series) or very close (in the case of the propylene glycol or isopropyl alcohol series), which suggests that the three-dimensional carbomer network is completely structured and the gel is entirely formed at both concentrations of the polymer (0.5% and 1%). The glycerol and propylene glycol formulations, loaded with drug liposomes, behaved similarly, for which the values of the Ostwald de Waele flow index were very close and <0.2. In contrast, the flow index calculated for the formulation based on 1% carbomer and isopropyl alcohol, loaded with liposomes, was slightly higher than that calculated for the liposomal hydrogel containing 0.5% polymer and the same cosolvent, but keeping the pseudoplastic character, n having low values (0.173 and 0.211, respectively) [50]. Comparing the control formulations with those loaded with liposomes from each series, there is a decrease in *n* values, indicating an increase in pseudo-plasticity, most likely due to a slight decrease in the strength of the polymer matrix in the presence of lipid vesicles.

The influence of the two formulation variables (carbomer concentration and cosolvent nature) was also indicated by the different values of the consistency index *K*, calculated for the analyzed formulations (loaded or not with drug liposomes). In all three series, there is a significant increase in the consistency index for the control formulations with the increase of the polymer concentration from 0.5% to 1%. The highest *K* values were obtained for the control formulations based on propylene glycol, at both polymer concentrations. Glycerol and especially isopropyl alcohol in the composition of the control formulations with 0.5% carbomer led to a decrease in the consistency index. Similar results were obtained for the 1% carbomer concentration, but the differences between the *K* values were smaller. On the other hand, the incorporation of liposomes in hydrogels based on 0.5% carbomer and glycerol, propylene glycol or isopropyl alcohol resulted in a drastic decrease in the consistency index of approximately 6 times, 10 times, and 5 times, highlighting both the influence of the type of cosolvent and lipid vesicles on this parameter of the Ostwald de Waele model.

The rheological properties of the analyzed hydrogels are modified by the incorporation of liposomes. Studies have shown that liposomes added in concentrations of 5 mg/mL or higher in carbomer-based hydrogels can affect their rheological characteristics, through properties such as lipid membrane stiffness (determined by the chemical composition of the lipid), the lipid concentration from formulation, and their surface charge [39,51,52]. It has been shown that the rheological properties (viscosity, consistency, thixotropy) of carbomer hydrogels can be modified by the addition of liposomes based on lipids with different chemical composition (phosphatidylcholine or hydrogenated phosphatidylcholine) in concentrations above 5 mg/mL. Additionally, the different rigidity of the liposome membrane was determined by the different chemical composition of the lipid used in the formulation (phosphatidylcholine forms membranes in the liquid state at room and body temperature, with a transition temperature of less than 0 °C, while hydrogenated phosphatidylcholine forms gel membranes, very rigid at normal temperature, with a transition temperature of about 50 °C) [39]. Another study showed that the incorporation of surface-free liposomes prepared with phosphatidylcholine or sodium phosphatidyl glycerol in 0.5% carbomer hydrogels leads to lower shear stress values with increasing deformation speed compared to those measured for the control hydrogel [51]. It was concluded that it was not the size of the liposomes but their composition that affected the rheological properties of the liposomal hydrogels. In addition, Boulmedarat et al. concluded that the incorporation of positively charged and sterically stabilized liposomes at a concentration of 2 mM lipid does not affect the rheological properties of some carbopol hydrogels, while their viscosity increases significantly in the presence of positively charged liposomes at a lipid concentration of 10 mM [52]. Thus, it can be considered that the differences between the rheological properties of liposomal hydrogels and those of control hydrogels containing 0.5% carbomer and analyzed in the present study can be attributed mainly to the increased rigidity of the liposomal membrane and the presence of a negative charge on the lipid vesicle surface. Phosphatidylcholine based liposomes are in the fluid state and deform slightly under the action of shear, to a small extent affecting the rheological characteristics of the hydrogel base in which they are incorporated. In contrast, the stiffness of liposomes containing saturated phospholipids is high at normal temperature and as a result, under the action of shear forces, they do not deform as easily as the hydrogel base, which causes a marked change in the rheological properties of the vehicle. The greater decrease in viscosity observed in the case of the liposomal hydrogel containing propylene glycol, compared to that of the liposomal formulation with glycerol or isopropyl alcohol can also be attributed to the cosolvent effect, which is added to the factors mentioned above. Propylene glycol is frequently used as a solvent for phospholipids, having the ability to increase the deformability of liposomal vesicles (elastic or ultra-deformable liposomes) [53]. As a result, it may be suggested that the propylene glycol in the hydrogel base composition may dissolve the phospholipid in the liposome structure, transferring it to the dispersion medium, which causes a further reduction in its polarity and thus a decrease in the viscosity/consistency of the hydrogel.

At the concentration of 1% carbomer, the hydrogel base is strongly structured, determining the rheological properties of the liposomal hydrogels and minimally influenced by the characteristics of the added liposomes.

Based on the results of the oscillation frequency sweep test it can be concluded that all studied hydrogels showed a predominantly elastic behavior resulting in mechanical and structural stability due to the presence of microstructure formed by a stable internal network of forces between the polymer and solvent chains.

It can be seen that all the hydrogels analyzed are non-Newtonian bodies with flow limits, but the value of this parameter varied, being influenced by the variables of the hydrogel base formulation and the presence of liposomal particles (Figure 13).

It is known that carbopol gels represent the archetype of yield stress fluids. These gels have a high elasticity and yield stress even at very low polymer concentrations. In an aqueous medium, carbopol macromolecules form by association microsponges which aggregate and swell by fixing the solvent molecules, due to the presence of ionizable groups (carbopol is a polyelectrolyte). Highly crosslinked systems do not swell as much as those with lower crosslinking. Swelling reduces the elasticity of the microgel particle but increases the volume fraction of the system, which affects its viscosity, elasticity, and yield stress. At a high-volume fraction, the microgels are compressed, and the viscosity increases more rapidly with the concentration above the critical value; the higher the crosslinking density, the more dramatic the effect. Microgels are unique in terms of rheological behavior due to their compressibility, which gives them increased elasticity and low rigidity, with the ability to avoid structural collapse under the action of shear, and they can deform considerably without compromising their configuration. Moreover, due to their compressibility, the gels have yield stress with a low degree of thixotropy.

The results of the analysis of the texture of the experimental hydrogels by the back compression-extrusion test are presented in Figure 14.

The concentration of the polymer, the cosolvent, and the loaded liposomes influence the textural characters of the studied hydrogels. As anticipated, the increase of the polymer concentration from 0.5% to 1% determined a linear increase of the values of the textural parameters of all analyzed hydrogels (hydrogel bases with Carbopol 940 and liposomal hydrogels) (Figure 13), as demonstrated by other previously published studies [45]. However, the degree of growth was different in the two series of hydrogels: slight increase (1.2–1.3 times) in the case of hydrogel bases and a considerable increase (3.9–7.9 times) in the case of liposomal hydrogels. More precisely, for the latter, the hardness and consistency (compressibility) increased about 4–6 times, and the cohesiveness and viscosity index increased about 5–8 times by doubling the polymer concentration. This means that an increase in carbopol concentration influences gel compressibility and hardness, thus affecting the ease of gel removal from the container. The high cohesiveness that arises from the increase of polymer concentration increases the performance of the product at the application site, allowing the complete restoration of the gel structure after administration. In addition, the higher values of the viscosity index of topical hydrogels with a higher concentration of the polymer will present with a prolonged adhesion at the site of administration (favorable to the therapeutic result). 

These results are consistent with those obtained by static and dynamic rheological tests (viscosity profiles, profiles of dynamic modules *G’* and *G’’,* and penetration values). In fact, in the case of control hydrogels, the three-dimensional networks created by the carbomer cross-chains are very stable for both polymer concentrations and represent an important obstacle for the introduction of the texture analyzer probe into the sample, thus increasing the strength required for this test step, for hardness, and consistency. However, the values of cohesiveness and viscosity index were slightly lower, indicating the lower force required to withdraw the probe of the apparatus from the sample. This can be correlated with lower structural integrity, with the thixotropy of control hydrogels (slightly more pronounced in those containing 1% polymer), which need more time to restore the structure after shearing. Similar behavior has been observed in other published studies [45,54].

Consequently, the differences of almost an order of magnitude between the values of the textural parameters of the liposomal hydrogels based on 0.5% and 1% Carbopol 940, respectively, can be attributed to the loaded liposomes rather than to the increase of the polymer concentration. There is a drastic decrease in the hardness, consistency, cohesiveness, and adhesiveness of hydrogels based on 0.5% carbomer with the addition of liposomes loaded with caffeic acid (Figure 14). It is clear that liposomal vesicles, by their composition and/or properties (except for their size), affect the strength of the bonds between polymer chains and the stability of the three-dimensional structure of gels with 0.5% carbomer, as evidenced by other previous studies [39,51]. However, Hurler et al. found that the texture of carbopol hydrogels does not change significantly with the addition of liposomal dispersions in concentrations of up to 15% [45]. Carbopol based hydrogels at 1% in the present study behaved similarly, their textural properties being slightly affected by the addition of liposomal vesicle dispersion, most likely due to the presence of a larger number of polymer chains capable of taking up the liquid. The data shown graphically in Figure 14 indicate the dependence of the texture of carbomer-based hydrogels on whether they are loaded/uncharged with cosolvent-type liposomes in the formulation. Comparing the hydrogel bases in terms of textural parameters, the glycerol formulations showed the highest hardness, consistency, cohesiveness, and adhesiveness, closely followed by the propylene glycol formulations, while the isopropyl alcohol formulations produced the lowest values of these parameters. These results can be attributed to the more pronounced structuring effect of glycerol than propylene glycol and isopropyl alcohol on the three-dimensional polymer network, which becomes harder, more consistent, more cohesive, and more adhesive, with greater resistance to probe movement in the samples. However, it is noted that the values of the textural parameters varied in a relatively narrow range, similar to that observed for the viscosity values, which confirms that the solvents used, although differing significantly in polarity, had only a side effect on the texture of the hydrogel bases, which was mainly dependent on the concentration of the carbomer. Comparing the data in Figure 13 which illustrates the effect of the addition of liposomes with caffeic acid on the texture of hydrogels based on carbomer and glycerol/propylene glycol/isopropyl alcohol, it is observed that in the 0.5% polymer formulations the isopropyl alcohol and glycerol (FA I and FG I) presented the highest values of hardness, consistency, cohesiveness and viscosity index. In contrast, for the propylene glycol (FP I) formulation, lower values of the textural parameters were obtained, which can be correlated with the lower stability of the three-dimensional structure of the gel containing the liposomal vesicles. This sensitivity can be attributed to the more pronounced effect of propylene glycol than glycerol and isopropyl alcohol in reducing the attraction between the polymer chains, affecting the rigidity of the three-dimensional structure of the gel. The liposomal formulation with propylene glycol from the hydrogel series with 1% Carbopol 940 (FP II), behaved in the same way, presenting the lowest values of the textural parameters. In contrast, in this series, a liposomal hydrogel with glycerol (FG II) demonstrated superiority in hardness, compressibility, cohesiveness, and adhesiveness over liposomal hydrogel with isopropyl alcohol. These results suggest that the internal structure of the 1% carbopol hydrogel and, thus, its texture are less affected by the incorporation of liposomes with caffeic acid in the presence of glycerol than isopropanol or propylene glycol.

The results of the penetrometric test reflected, as in the case of the rheological tests described above (viscometrical test and dynamic amplitude and frequency sweep tests), the significant differences in consistency (hardness) between the experimental preparations. Responsible for these differences are the vehicle formulation variables (polymer concentration and cosolvent type) and the presence of liposomal particles. 

Consequently, the stretching capacity decreased with increasing carbomer concentration, both in the case of hydrogel bases and those loaded with liposomal vesicles. Similar to the back compression-extrusion test, the dependence on the two cosolvent type parameters in the formulation is observed. For the same concentration of carbomer, and for the highest values of firmness and shear work, the lowest display capacities were shown by the hydrogels with isopropyl alcohol, followed by those with glycerol. In contrast, propylene glycol hydrogels showed the lowest firmness and resistance to destructuring, indicating an increased display capacity.

## 4. Materials and Methods

### 4.1. Materials

Glycerin, carbopol, isopropyl alcohol, propylene glycol, ethyl alcohol, triethanolamine were purchased from Farmachim, Ploiesti, Romania. All substances used have purity according to the analysis bulletins issued by the manufacturer. Ethanol—Farmachim 10 SRL, Romania, lot 9110086696, triethanolamine—Merck KgaA, Germany, lot K49471379, Glycerol—Nordische Oelwerke-Walther Carroux GmbH & Co. KG series 67340, isopropyl alcohol lot 1859097, Eco-Mold Invest SRL Romania, 1,2-propylene glycol USP, lot. 3683, Elton Corporation SA Romania. The carbopol used is the apron—Synthalen K—batch.no. 0219L01 (3V Sigma Manufacturing) supplied by Vitamar, Bucharest, Romania, of pharmaceutical or analytical purity and was used as such. The distilled water was obtained in the laboratory.

### 4.2. Encapsulation of Liposomes with Caffeic Acid in Hydrogels

In our studies we followed the entrapment of caffeic acid in liposomes, so six liposomal formulas were prepared, and after the morphological and structural characterization, we selected two formulas (DPPC-50 and SC-50) that trapped and yielded caffeic acid to ensure sufficient bioavailability for local application. We mention that the preparation of liposomes was performed by the thin film method [55]. The substances and quantities used to obtain liposomes are given in Table 6.

Hydrogels were obtained by dispersing carbopol in a mixture of solvent and water. Stirring was continued with a mixer for better dispersion. The mixture was allowed to stand for 2 hours at room temperature. When the process of hydrating the carbopol was completed, triethanolamine dispersed in a mixture of ethyl alcohol and water was added in small amounts and under continuous stirring until the gel was clarified (Figure 15). The composition of the three gel bases is shown in Table 6.

Cosolvents were used to give the preparation a good viscosity and a translucent appearance [56,57].

Triethanolamine is an agent for increasing the solubility of the active substance in a vehicle. It has also been used to neutralize the free carboxylic acid groups in carbopol to pH 6 ± 0.5 [58].

Carbopol is a crosslinked polyacrylic acid polymer extremely effective in modifying rheological properties, able to provide a high viscosity forming clear gels. Ethyl alcohol has been used to promote the dispersion of macromolecules in water. Glycerin, isopropyl alcohol, and propylene glycol act as cosolvents.

We obtained 6 hydrogels in which we incorporated liposomes with caffeic acid and 6 hydrogels without liposomes.

### 4.3. Determination of Macroscopic Characteristics and pH

The formulation of the gels was followed by their macroscopic evaluation. Organoleptic properties such as color, odor, physical appearance, and homogeneity were assessed by visual perception immediately after preparation and after 3 months being stored at room temperature (20 ± 1 °C) [59].

The pH was determined potentiometrically, according to Romanian Pharmacopoeia [60], using a portable digital pH meter (Sension™ 1, Hach Company, Loveland, CO, USA). More than 5 g hydrogel was added to 20 mL of distilled water previously heated to 37 ± 2 °C and stirred vigorously for 1 min. After cooling, the dispersion was filtered, and the pH was determined in the filtrate. Each determination was made in triplicate.

### 4.4. Viscoelastic Measurements of Liposomal Hydrogels

The rheological properties of experimental hydrogels were studied by static and dynamic tests, using a stress-controlled rheometer (RheoStress 1, HAAKE, France). The Peltier module (TCP/P) provided temperature control (23 °C) during rheological tests. All determinations were performed using a con-plate geometry (cone diameter was 60 mm and cone angle 1°), which operated with an aperture of 0.052 mm.

#### 4.4.1. The Steady-State Flow Test

The test was performed in rotation mode, varying the plate shear rate (0.05–100.0 1/s) in ascending and descending cycles, 120 seconds each, with the simultaneous measurement of the shear stress exerted on the cone. By graphically representing the shear stress and viscosity according to the shear rate, rheograms and viscosity curves of hydrogels were obtained. HAAKE RheoWin Data Manager 4 version 4.3 was used to analyze and process flow and viscosity data by fitting with the following rheological models: Ostwald de Waele (Equations (1) and (2)), Herschel–Bulkley (Equations (3) and (4)), model Bingham (Equation (5)), and the Casson model (Equation (6)):(1)$$\tau =K\cdot {\dot{\gamma }}^{n}$$
(2)$$\eta =K\cdot {\dot{\gamma }}^{n-1}$$
(3)$$\tau =\tau_{0}+K\cdot {\dot{\gamma }}^{n}$$
(4)$$\eta =\eta_{0}+\tau_{0}/\dot{\gamma }$$
(5)$$\tau =\tau_{0}+K\cdot {\dot{\gamma }}^{n}$$
(6)$$\tau =\sqrt[{n}]{\tau_{0}^{n}+{\left(\eta_{0}\cdot \dot{\gamma }\right)}^{n}}$$
where: *τ* = shear stress (Pa), *K* = consistency index (Pa ∙ s), *γ* = shear rate; *n* = flow index; *η* = viscosity and *τ_0_* = flow point (Pa). For 0 < *n* <1, it had a pseudoplastic behavior (thin to shear); the lower its values, the more pseudoplastic the system is. The accuracy of the fit was assessed based on the value of the calculated correlation coefficient of the model.

Thixotropy was quantified by the thixotropy index (relative hysteresis area, *A_R_*, %) [61], which is the percentage of the area destroyed by shear and can be expressed by the Equation:$$A_{R}=\frac{\left(A_{UP}-A_{Down}\right)}{A_{UP}}\times 100$$
where *A_UP_* is the area under the upward flow curve, and *A_Down_* is the area below the downward flow curve. Systems with A*_R_* values above 5% can be considered thixotropic, and those with *A_R_* below 5% show only pseudoplastic behavior (shear-thinning) without thixotropy [61].

#### 4.4.2. Dynamic Tests (Oscillators)

The viscoelasticity of the hydrogels was assessed by the following tests: the oscillation amplitude sweep test and the oscillation frequency sweep test. The oscillation amplitude sweep test was performed in oscillation mode, at a frequency of 1 Hz on different ranges of shear stress (0.5–10 Pa, 0.5–100 Pa, 0.5–500 Pa). This test allowed the determination of the linear viscoelasticity region of the samples and of the viscoelastic modules (accumulation modulus, and loss modulus, *G’,* and loss modulus *G’’*) as a function of the shear stress, τ. In the oscillation frequency sweep test, the two modules and the complex viscosity (η*) [38] were recorded over a wide frequency range (0.05–50 Hz), keeping the shear stress constant (5.0 Pa), its value being selected from the linear viscoelasticity region.

Each experiment was performed in triplicate.

#### 4.4.3. The Yield Stress

The yield stress (τ_0_) of the experimental hydrogels was determined directly, by the oscillation amplitude sweep test, after determining their viscoelastic linear region. The stress-controlled rheometer (RheoStress 1, HAAKE, France) was used to perform amplitude scans in the range of 1–500 Pa at a constant frequency of 1 Hz, allowing the generation of *G’* and *G’’* dynamic module curves as a function of the shear stress. The flow threshold is the critical stress at which an irreversible plastic deformation occurs, indicated by the decrease of the value of the elastic modulus G’.

#### 4.4.4. Texture Analysis of Hydrogels

The textural properties of the hydrogel formulations were evaluated by the back extrusion compression test using a texture analyzer (TAXT Plus, Stable Micro Systems, London, UK) equipped with a 5 kg load cell and a probe extrusion (a compression disc with a diameter of 35 mm, attached to a rod). All experiments were performed at room temperature, in compression mode, by testing each hydrogel in triplicate, and the results were presented as average ± standard deviation (SD).

50 g of the sample was placed in a 100 mL container up to a height of 50 mm, avoiding the incorporation of air. The back extrusion compression test of the sample was performed in the following steps: (1) the extrusion probe was lowered at a speed of 5 mm/s to the surface of the sample; (2) a trigger force (5 g) was applied to detect contact with the sample, then the probe continued to descend at a speed of 2 mm/s, compressing the hydrogel sample to a depth of 10 mm; (3) the probe returned to the surface of the hydrogel at the same speed (2 mm/s), then was raised at a speed of 5 mm/s to the initial position. Data acquisition and force-time curve tracing were performed using the instrument software (Texture EXPONENT version 3.0.5.0). Based on the force-time curve, the texture parameters were calculated [62,63]: firmness (in g) as the maximum force required for the deformation of the sample (corresponding to the maximum value of the compression depth of the probe); consistency (in g·s) as the area of the positive region corresponding to the total work required for compression; cohesiveness (in g) as the maximum force required to withdraw the probe from the sample to the surface, after compression of the sample; viscosity index (in g·s) as the area of the negative region.

#### 4.4.5. Determination of Consistency

The consistency of experimental hydrogels was also determined by the official penetrometric method [64]. The test was performed at 25 ± 0.5 °C, according to the procedure and under the conditions described in the pharmacopeia, using a penetrometer (PNR 12, Petrolab, Germany), equipped with a micro-cone (as a penetration accessory) and a suitable container. Each experiment was performed in triplicate.

The results obtained, namely the values of the degree of penetration (in mm), were used to express the consistency of the hydrogels. High penetration values indicate a lower consistency.

#### 4.4.6. Spreadability Capacity Determination

The spreadability capacity of the experimental hydrogels was determined by two different methods: the parallel plate method and the method based on texture analysis.

For the parallel plate method, the Pozo Ojeda-Sune Arbussa extensiometer was used and the working technique followed was the one described in the literature [65,66].

To test the hydrogel spreadability capacity using the texture analyzer (TAXT Plus, Stable Micro Systems, London, UK), a specific accessory (TTC Spreadability Rig HDP/SR, Stable Micro Systems, London, UK) was used. During the test, the conical probe was lowered over a distance of 23 mm at a speed of 3 mm/s to the surface of the sample in the support, then penetrated the sample, forcing it to flow outwards at 45° between the surfaces of the two cones (male and female). The force (in g) required for the penetration of the male-type conical probe into the sample from the female-type conical support was measured. From the force versus time curve, two parameters were selected as indicators of the firmness and spread capacity of the hydrogels: the maximum force value and the shear work, respectively (the area under the curve in the region with positive *f* force values). Lower force values indicate better spread capacity. Four samples from each hydrogel formulation were tested at room temperature. Data analysis was performed using Exponent software ver. 6.1.18.0 (Stable Micro Systems, London, UK). Each experiment was performed in triplicate.

### 4.5. Release of Caffeic Acid from Hydrogels

A system of six Franz diffusion cells (Microette-Hanson system, model 57-6AS9, Copley Scientific Ltd., Nottingham, UK) was used to evaluate CA release from hydrogels. The receptor chamber has a diffusion area of 1767 cm^2^ and a volume of 6.5 mL and each diffusion cell was filled with phosphate buffer (pH 7.4) mixed with freshly prepared 30% ethanol. Synthetic polysulfone membranes with a diameter of 25 mm and a pore size of 0.45 m—Tuffryn^®^, PALL Life Sciences HT-450, batch T72556, were hydrated by immersion in a receiving medium for 30 minutes before use, then mounted between the donor and acceptor compartment of the Franz diffusion cell. An amount of about 0.500 g of hydrogel was brought into the diffusion cell capsule. The system was maintained at 32 ± 1 °C and the receiving medium was stirred continuously (600 rpm) using a magnetic stirrer to avoid the effects of the diffusion layer. 0.5 mL of receptor solution was taken at various time intervals (30 min, 1, 2, 3, 4, 5, 6, 7, 8, 12 h) and replaced with a fresh receptor medium to maintain a constant volume (6.5 mL) during the test. The amount of CA released was determined using a UV-VIS spectrophotometric method; the reading is performed at 325 nm.

The studies performed should be useful for evaluating the topical performance of topical formulations containing CA.

## 5. Conclusions

The detailed analysis of the stationary shear rheometry on different flow parameters was performed by highlighting the influence of formulation factors, notably the presence of liposomes entrapped with caffeic acid on the rheological behavior of carbopol hydrogels.

Because rheology has an important role both technologically and biopharmaceutically, we believe that the study shows the correlation of rheological results with the basic composition of hydrogels and kinetic characteristics, as required for the optimization of topical pharmaceuticals.

The comparative analysis of the firmness and shear stress recorded for the hydrogel bases and for the liposomal hydrogels shows that the latter revealed a higher display capacity of the vehicles, indicated by the much lower values of firmness and shear work. It may be suggested that the addition of liposomes with caffeic acid affects the strength of the adhesion/interaction forces between the components of the three-dimensional network of carbomer-based hydrogels. The results of this test are consistent with those obtained by the compression-extrusion test and then by rheological tests. The rheological analysis applied to carbopol gels and liposomes with CA can serve as a model for hydrogels with various application sites.

## Figures and Tables

**Figure 1 pharmaceuticals-15-00175-f001:**
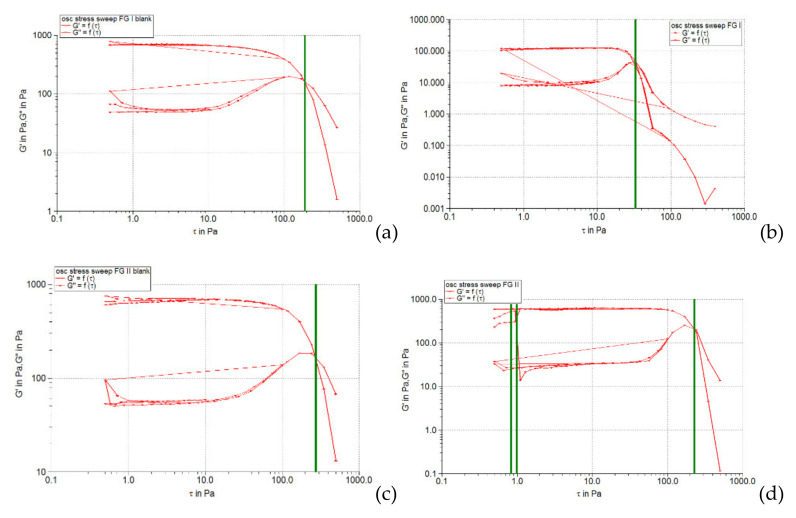
The behavior of viscoelastic modules *G’* and *G’’* depending on the shear stress for hydrogels based on Carbopol 940 and glycerol, simple or loaded with caffeic acid liposomes: (**a**) FGI blank, (**b**) FGI, (**c**) FGII blank, (**d**) FGII.

**Figure 2 pharmaceuticals-15-00175-f002:**
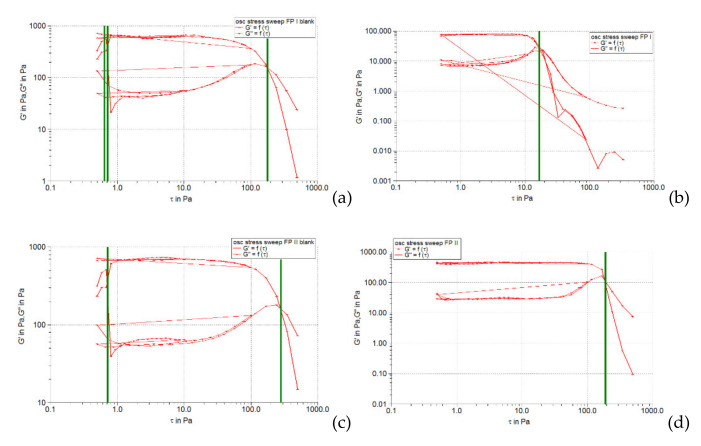
The behavior of viscoelastic modules *G’* and *G’’* depending on the shear stress for hydrogels based on Carbopol 940 and propylene glycol, simple or loaded with caffeic acid liposomes: (**a**) FPI blank, (**b**) FPI, (**c**) FPII blank, (**d**) FPII.

**Figure 3 pharmaceuticals-15-00175-f003:**
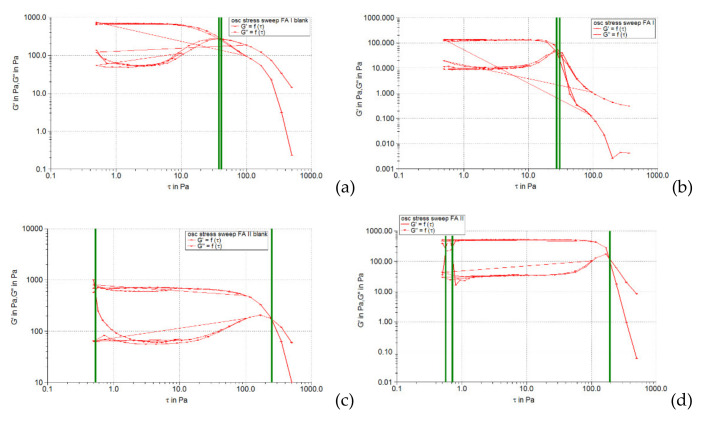
The behavior of viscoelastic modules *G’* and *G’’* depending on the shear stress for hydrogels based on Carbopol 940 and isopropyl alcohol, simple or loaded with caffeic acid liposomes: (**a**) FAI blank, (**b**) FAI, (**c**) FAII blank, (**d**) FAII.

**Figure 4 pharmaceuticals-15-00175-f004:**
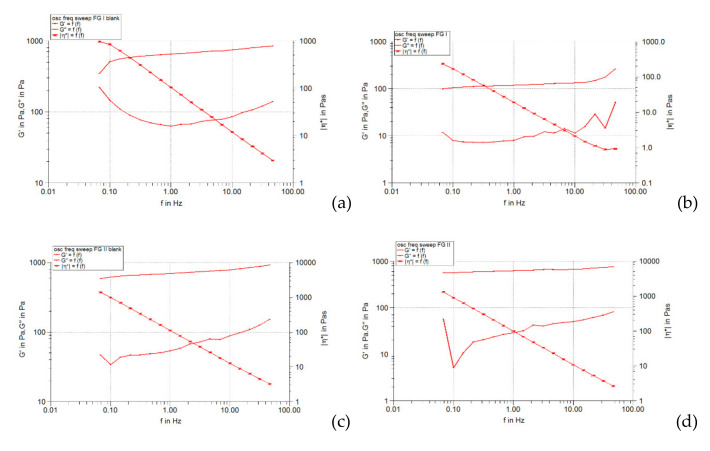
Behavior of *G’* and *G’’* viscoelastic modules depending on frequency for Carbopol 940 and glycerol hydrogels, simple or loaded with caffeic acid liposomes: (**a**) FGI blank, (**b**) FGI, (**c**) FGII blank, (**d**) FGII.

**Figure 5 pharmaceuticals-15-00175-f005:**
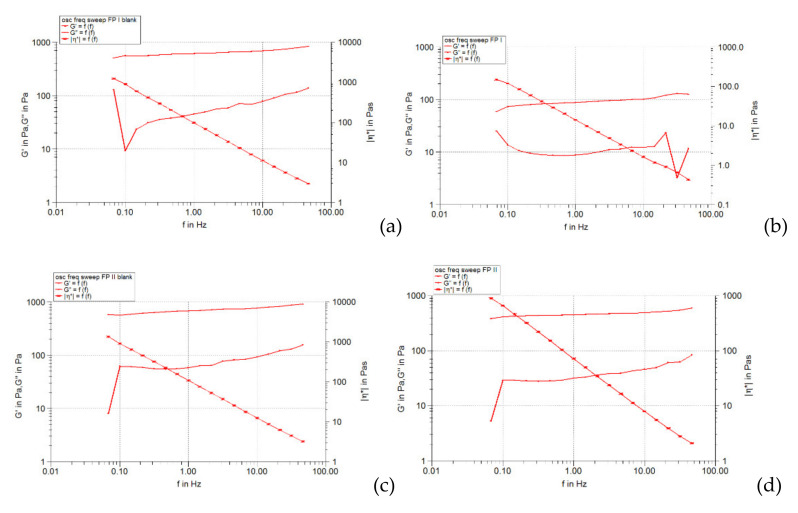
Behavior of *G’* and *G’’* viscoelastic modules depending on frequency for Carbopol 940 and propylene glycol hydrogels, simple or loaded with caffeic acid liposomes: (**a**) FPI blank, (**b**) FPI, (**c**) FPII blank, (**d**) FPII.

**Figure 6 pharmaceuticals-15-00175-f006:**
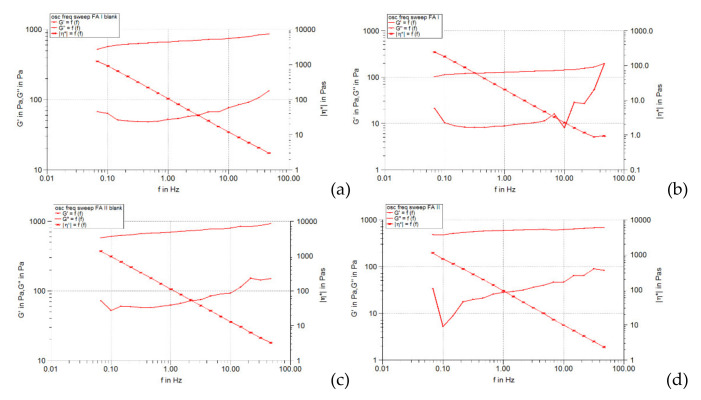
The behavior of *G’* and *G’’* viscoelastic modules depending on frequency for Carbopol 940 and isopropyl alcohol, simple or loaded with caffeic acid liposomes: (**a**) FAI blank, (**b**) FAI, (**c**) FAII blank, (**d**) FAII.

**Figure 7 pharmaceuticals-15-00175-f007:**
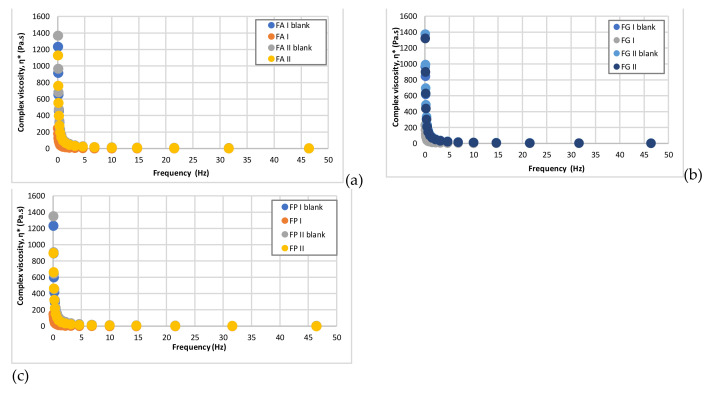
Variation of complex viscosity η* as a function of frequency for the studied hydrogels: (**a**) FAI blank, FAI, FAII blank, FAII; (**b**) FGI blank, FGI, FGII blank, FGII; (**c**) FPI blank, FPI, FPII blank, FPII.

**Figure 8 pharmaceuticals-15-00175-f008:**
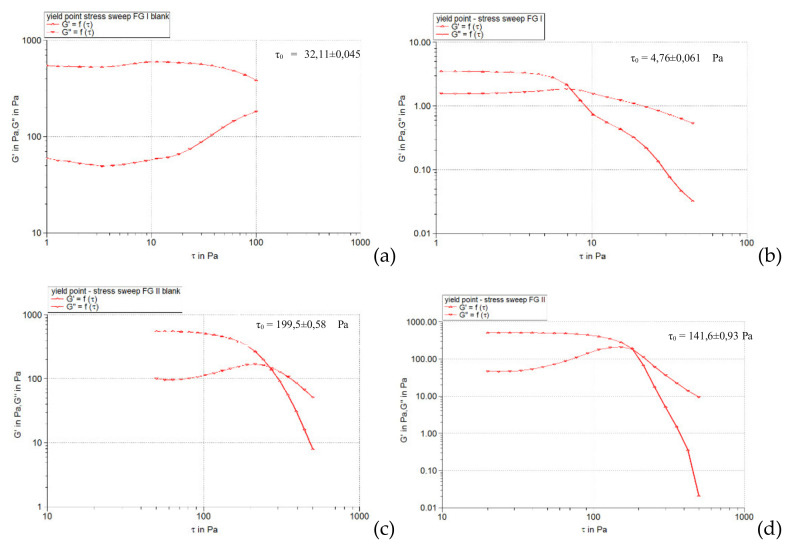
*G’* and *G’’* modulus curves as a function of stress at 1 Hz for Carbopol 940 and glycerol-based hydrogels, simple or loaded with caffeic acid liposomes: (**a**) FGI blank, (**b**) FGI, (**c**) FGII blank, (**d**) FGII.

**Figure 9 pharmaceuticals-15-00175-f009:**
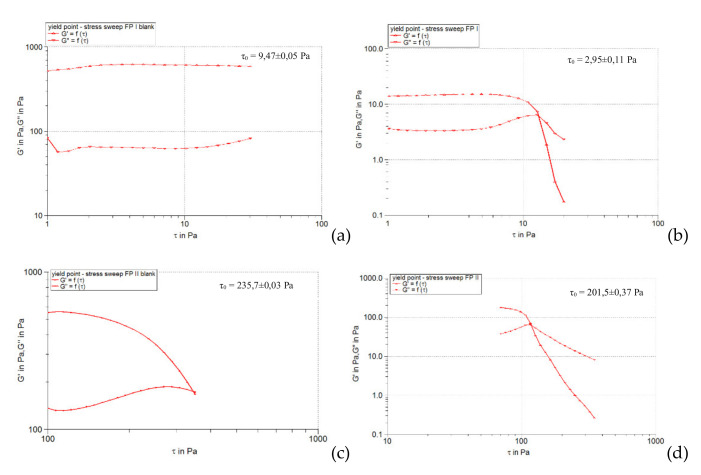
*G’* and *G’’* modulus curves as a function of stress at 1 Hz for Carbopol 940 and propylene glycol-based hydrogels, simple or loaded with caffeic acid liposomes: (**a**) FPI blank, (**b**) FPI, (**c**) FPII blank, (**d**) FPII.

**Figure 10 pharmaceuticals-15-00175-f010:**
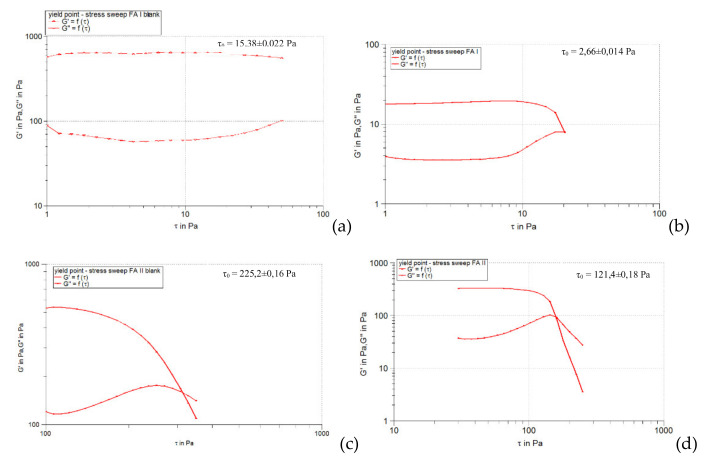
*G’* and *G’’* modulus curves as a function of stress at 1 Hz for hydrogels based on Carbopol 940 and isopropyl alcohol, simple or loaded with caffeic acid liposomes: (**a**) FAI blank, (**b**) FAI, (**c**) FAII blank, (**d**) FAII.

**Figure 11 pharmaceuticals-15-00175-f011:**
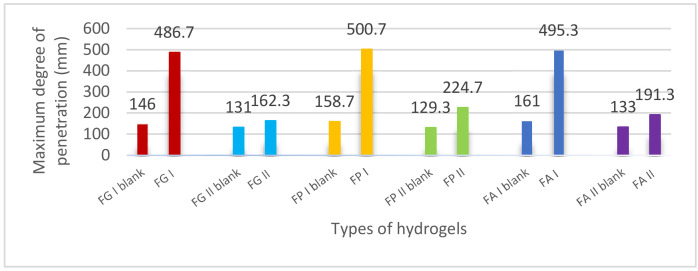
The influence of the formulation composition on the consistency of the studied experimental hydrogels.

**Figure 12 pharmaceuticals-15-00175-f012:**
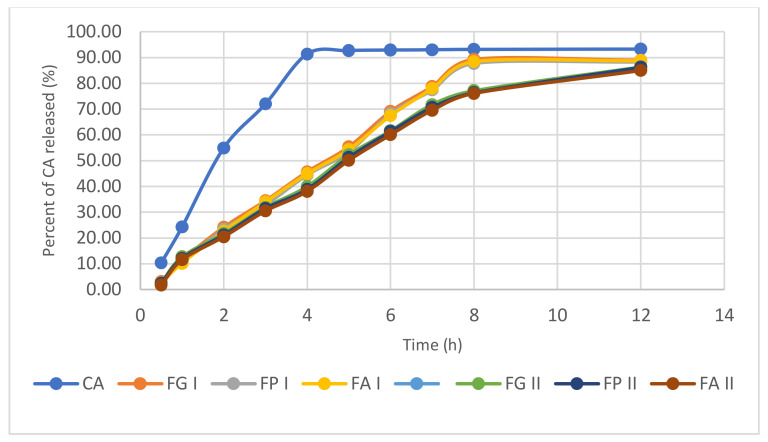
The percent of CA released in time (h) from hydrogels.

**Figure 13 pharmaceuticals-15-00175-f013:**
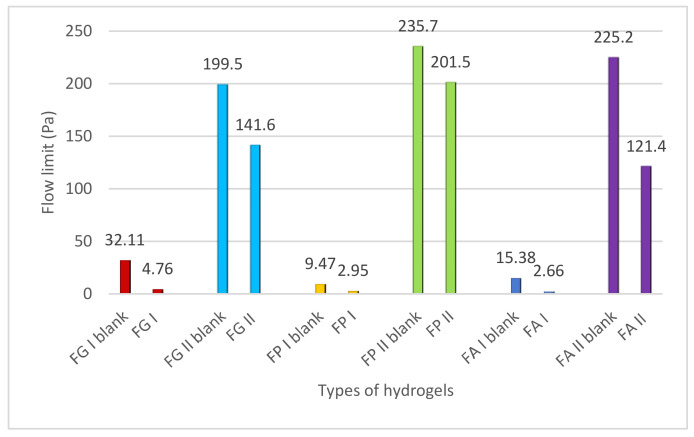
Influence of formulation variables on the yield stress of the studied hydrogels.

**Figure 14 pharmaceuticals-15-00175-f014:**
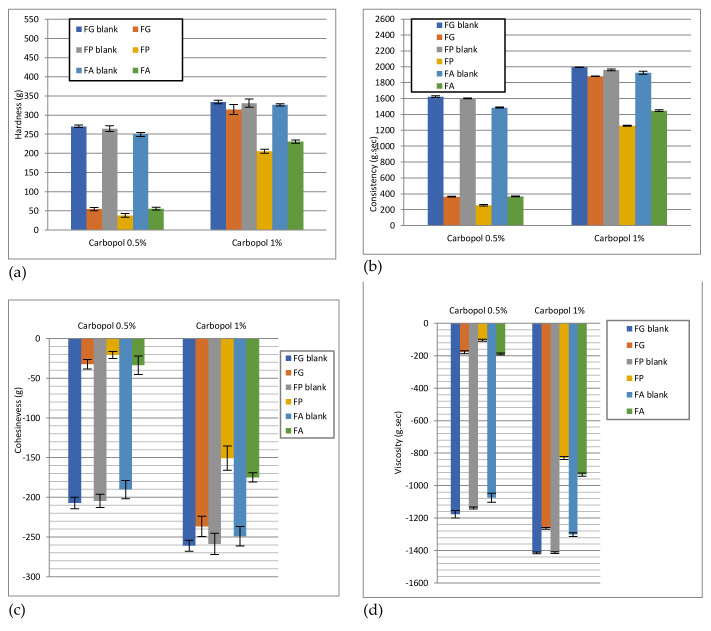
Textural parameters of Carbopol 940 based hydrogels determined in the back compression-extrusion test: hardness (**a**), consistency (**b**), cohesiveness (**c**), and viscosity index (**d**). Values are expressed as mean ± standard deviation (*n* = 3).

**Figure 15 pharmaceuticals-15-00175-f015:**
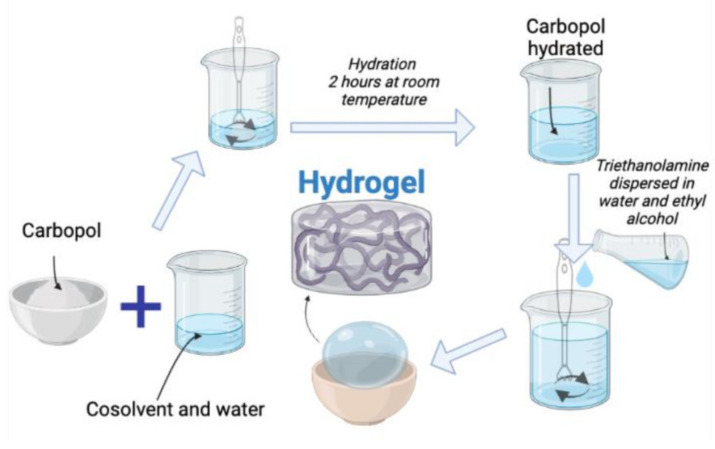
Hydrogel’s preparation graphic abstract.

**Table 1 pharmaceuticals-15-00175-t001:** Quantities of substances used for the preparation of liposomal hydrogels.

Sample	FG I (g)	FP I (g)	FA I (g)	FG II (g)	FP II (g)	FA II (g)	FG I Blank (g)	FP I Blank (g)	FA I Blank (g)	FG II Blank (g)	FP II Blank (g)	FA II Blank (g)
Liposome DPPC-50	1	1	1	-	-	-	-	-	-	-	-	-
Liposome SC-50	-	-	-	1	1	1	-	-	-	-	-	-
Carbopol 940	0.50	0.50	0.50	1	1	1	0.50	0.50	0.50	1	1	1
Glycerin	20	-	-	20	-	-	20	-	-	20	-	-
Isopropyl alcohol	-	-	20	-	-	20	-	-	20	-	-	20
Propylene glycol	-	20	-	-	20	-	-	20	-	-	20	-
Ethyl alcohol 96°	10	10	10	10	10	10	10	10	10	10	10	10
Triethanolamine	0.50	0.50	0.50	1	1	1	0.50	0.50	0.50	1	1	1
Distillate water	68	68	68	67	67	67	69	69	69	68	68	68

**Table 2 pharmaceuticals-15-00175-t002:** Regression parameters of rheological models (Ostwald de Waele and Herschel–Bulkley) and rheological parameters of Carbopol 940-based hydrogels were obtained by fitting the ascending portion of the rheograms with the respective models.

Formulation Code	Correlation Coefficient (R) Model		Parameters Model Ostwald de Waele		Parameters Model Herschel–Bulkley		
	Ostwald de Waele	Herschel–Bulkley	K	n	K	n	τ_0_ (Pa)
FG I blank	0.9964	0.9967	135.41	0.239	146.3	0.229	−12.35
FG I	0.9644	0.9817	23.06	0.173	118.0	0.044	−94.33
FG II blank	0.9926	0.9946	178.30	0.239	264.5	0.188	−94.57
FG II	0.9823	0.9940	149.91	0.183	377.0	0.086	−219.10
FP I blank	0.9948	0.9981	139.70	0.229	208.0	0.174	−67.76
FP I	0.9790	0.9934	14.24	0.197	46.6	0.079	−32.67
FP II blank	0.9942	0.9984	186.11	0.235	296.1	0.171	−109.50
FP II	0.9806	0.9950	115.31	0.191	464.4	0.059	−341.71
FA I blank	0.9952	0.9969	107.60	0.261	137.5	0.226	−33.13
FA I	0.9709	0.9934	20.54	0.173	137.8	0.034	−116.80
FA II blank	0.9943	0.9977	183.32	0.224	267.4	0.172	−83.21
FA II	0.9880	0.9944	131.32	0.211	276.5	0.119	−140.42

**Table 3 pharmaceuticals-15-00175-t003:** Apparent viscosity and thixotropy index of experimental hydrogels.

Formulation Code	Apparent Viscosity (Pa.s)	Thixotropy Index (%)
FG I blank	3.990	36.41
FG I	0.518	6.82
FG II blank	5.350	34.34
FG II	3.429	32.34
FP I blank	3.956	29.67
FP I	0.353	5.24
FP II blank	5.327	76.60
FP II	2.806	31.27
FA I blank	3.504	26.49
FA I	0.454	7.43
FA II blank	4.914	78.42
FA II	3.464	42.84

**Table 4 pharmaceuticals-15-00175-t004:** The results of the amplitude sweep test for the studied liposomal hydrogels.

Formulation Code	*G’* (Pa)	*G’’* (Pa)
FG I blank	697.66 ± 8.11	53.42 ± 3.02
FG I	119.16 ± 1.61	9.21 ± 1.41
FG II blank	682.44 ± 18.18	55.57 ± 1.73
FG II	607.56 ± 13.20	30.18 ± 2.17
FP I blank	614.73 ± 15.31	47.30 ± 4.25
FP I	78.77 ± 0.85	8.05 ± 1.09
FP II blank	699.89 ± 12.35	59.24 ± 4.44
FP II	450.99 ± 3.96	29.59 ± 0.57
FA I blank	673.53 ± 9.53	55.33 ± 2.78
FA I	128.17 ± 3.46	9.93 ± 0.86
FA II blank	669.35 ± 48.24	61.69 ± 2.98
FA II	517.42 ± 3.38	34.09 ± 0.49

**Table 5 pharmaceuticals-15-00175-t005:** Parameter values measured during testing the ability to display experimental hydrogels with the texture analyzer.

Formulation Code	Firmness (g)	Mechanical Shear Work (g·s)
FG I blank	1764.23 ± 12.30	1156.10 ± 32.55
FG I	46.21 ± 5.37	48.06 ± 20.16
FG II blank	2986.66 ± 18.90	1615.13 ± 26.36
FG II	1207.20 ± 7.82	946.85 ± 12.04
FP I blank	1216.80 ± 11.37	931.40 ± 32.22
FP I	927.43 ± 13.58	237.12 ± 11.57
FP II blank	1344.75 ± 13.48	979.64 ± 32.39
FP II	600.40 ± 20.57	436.52 ± 16.97
FA I blank	3912.00 ± 11.46	1896.40 ± 13.03
FA I	149.53 ± 14.38	105.91 ± 17.37
FA II blank	4273.60 ± 22.99	2314.30 ± 12.28
FA II	1526.10 ± 24.97	855.35 ± 11.01

**Table 6 pharmaceuticals-15-00175-t006:** Quantities of substances used for the preparation of liposomes with caffeic acid.

Amount of Substances (mg)	Type of Liposomes	
	DPPC-50	SC-50
Caffeic acid (CA)	50	50
Phosphatidylcholine (PC)	50	80
1,2-dipalmitoyl-sn-glycero-3-phosphocholine (DP-PD)	50	-
Sodium cholate (SC)	-	20
Cholesterol (CHL)	2.5	2.5

CA—caffeic acid, CHL—cholesterol, SC—sodium cholate, PC—phosphatidylcholine, DP-PD—1,2-dipalmitoyl-sn-glycero-3-phosphocholine.

## Data Availability

The data presented in this study are available in the article and Appendix A.

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
