# Peer review of "Study for Evaluation of Hydrogels after the Incorporation of Liposomes Embedded with Caffeic Acid"

_pharmaceuticals, 2022, doi:10.3390/ph15020175_

Round 1
Reviewer 1 Report
In the manuscript entitled "STUDY FOR THE PHARMACEUTICAL EVALUATION OF HYDROGELS AFTER THE INCORPORATION OF LIPOSOMES EMBEDDED WITH CAFFEIC ACID", two different formulas of caffeic acid liposomes were incorporated into three different formulas of carbopol-based hydrogels and analyzed for characterization. Based on the evaluation data, the authors concluded that the textural properties of 1% Carbopol-based hydrogels were slightly affected by the addition of liposomal vesicle dispersion and the firmness and shear work increased with increasing carbomer concentration. The data of this manuscript provide a reference for the development of caffeic acid agents and the study of carbomer hydrogel properties. The data presentation is difficult to discern. The data should be further analyzed and presented concisely and clearly.
Reviewer 2 Report
In this work, some formulations with and without liposomes embedded with caffeic acid and carbopol-based hydrogels were studied. Rheological and textural test were carried out and, also, the release of caffeic acid was evaluated.
In my opinion, it is an interesting work for publication in Pharmaceuticals.
Nevertheless, some considerations could be made:
Title
I think it seems redundant “Study for the pharmaceutical evaluation…”
Abstract
In my opinion, the words “Results” and “conclusions” should be deleted from de Abstract
Results
I suggest changing the order of the epigraphs.
I think that 2.2 The release studies should be placed at the end of the paper, after the global characterization of the formulas
Figure 2. I cannot clearly see the 6 graphs compiled.
Table 3 should be Table 1 because it is shown in the first place. Please, delete “g”.
Thank you
Reviewer 3 Report
This paper presents transverse analysis between 12 types of hydrogels that carries CA cargoes or not, detailed characterization including release behavior, PH and viscoelastic measurement were shown. However, the novelty and practical significance of this lengthy paper are kind of unsatisfying. Besides, the structure of manuscript is really confusing. Specifically:
- The introduction part is far too tedious, but the key subject hydrogels are barely mentioned;
- Explantation of FG/FP/FA hydrogels are in Part. 4, which is particularly cunfusing while reading the previous sections;
- Why these kind of hydrogels were chosen since there are thousands of hydrogels available, the reasons should be listed;
- Since CA hold theurapeutic potiential in multiple areas, release experiments in the same environment seem to be useless;
- The section 2.3 seems to be total worthless, a single photograph might be more helpful.
Reviewer 4 Report
The article - 1527890: STUDY FOR THE PHARMACEUTICAL EVALUATION OF HYDROGELS AFTER THE INCORPORATION OF LIPOSOMES EMBEDDED WITH CAFFEIC ACID was evaluated for publication in Pharmaceuticals and proposed for publication after a minor revision of the paper.
The abstract is too general. It is suggested that the authors restructure the abstract without subsections for methods, results and conclusions.
The introduction (first and second paragraphs) lacks relevant and new references at CA. It is suggested to include some new references, such as: ISLAMČEVIĆ R.M. et al.: Validated stability-indicating GC-MS method for characterization of forced degradation products of trans-caffeic acid and trans-ferulic acid. Molecules. 23 April 2021, vol. 26, no. 9, p. 1-12 and ISLAMČEVIĆ R.M. et al. Choline chloride based natural deep eutectic solvents as extraction media for extracting phenolic compounds from chokeberry (Aronia melanocarpa). Molecules. Apr. 2020, vol. 25, iss. 7, p. 1-14.
The numbering of all the chapters in number 2 and especially in 2.4 should be reviewed.
There are too many images in the work and it is suggested that some images be moved to the supplementary material.
All results are given to 2 decimal places. Why and how was this evaluated using SD?
Also the conclusions are very general and should be more specific.
Add new references as mentioned above.
Reviewer 5 Report
Dear authors:
1- table 2 is very small, can you implement the data in the text?
2-Can you provide higher quality figures, they are unreadable
3-line 597-599 uptown 603 in the discussion section must be combined and discussed more as a paragraph
4-Can you change Fig 17 to a 2D one?
5- I cannot read Figure 18, many figures in the paper are not clear
6- Line 753 ~ change to graphical abstract, (is it authentic figure?)
7-"The studies performed should be useful for evaluating the topical performance of 914 topical formulations containing CA." should be removed from the conclusion and discussed in the discussion section
Author Response
Esteemed reviewer,
Thank you for your comments.
1- table 2 is very small, can you implement the data in the text?
We deleted that table.
2-Can you provide higher quality figures, they are unreadable
The observations were taken into account. We changed.
3-line 597-599 uptown 603 in the discussion section must be combined and discussed more as a paragraph
The observations were taken into account. We added the paragraph.
4-Can you change Fig 17 to a 2D one?
We have modified fig. 17.
5- I cannot read Figure 18, many figures in the paper are not clear
We have modified.
6- Line 753 ~ change to graphical abstract, (is it authentic figure?)
It is an authentic figure make by o special program (BioRender.com).
7-"The studies performed should be useful for evaluating the topical performance of 914 topical formulations containing CA." should be removed from the conclusion and discussed in the discussion section
We have modified.
Respectfully
Laura Vicas et al
Reviewer 6 Report
Diffusion study....How much amount of drug taken for release study.
Give the solubilty data of caffiec acid.
How much solubility of caffiec acid in water. What is the need to prepare liposomes based gel.
Give the particle size, PDI, ZP and TEm images.
The manuscript is clearly presented. There are number of editing shown in the file.
THe text in images are not clear.
The release data must be added with stats.
The release study must be upto 12 h.
Author Response
Esteemed reviewer,
Thank you for your comments.
Diffusion study.... How much amount of drug taken for release study.
We added in manuscript.
Give the solubility data of caffeic acid.
We added in manuscript
How much solubility of caffeic acid in water. What is the need to prepare liposomes based gel.
We added in manuscript
Give the particle size, PDI, ZP and TEM images.
These determinations are part of another paper, under review, which describes how to prepare liposomes, determine the efficiency of entrapment, characterization of liposomes (by AFM, DLS, zeta potential), determination of CA release from liposomes, and experiments on carcinoma cell line.
The manuscript is clearly presented. There are number of editing shown in the file.
Thank you very much for your appreciation.
The text in images are not clear.
We changed
The release data must be added with stats.
We changed.
Table I. The percent of CA released in time (h) from hydrogels, expressed as mean ± standard deviation
|
Time (h) |
CA (%) |
FG I (%) |
FP I (%) |
FA I (%) |
FG II (%) |
FP II (%) |
FA II (%) |
|
0.5 |
10.32±0.21 |
3.05±0.13 |
2.44±0.09 |
2.64±0.08 |
2.50±0.11 |
2.45±0.09 |
1.65±0.11 |
|
1 |
24.24±0.11 |
11.29±0.22 |
10.38±0.24 |
10.12±0.29 |
12.66±0.26 |
12.06±0.20 |
11.51±0.09 |
|
2 |
54.88±0.18 |
24.16±0.41 |
23.35±0.11 |
22.58±0.41 |
21.72±0.38 |
21.12±0.30 |
20.42±0.037 |
|
3 |
71.96±1.25 |
34.38±0.54 |
32.76±0.33 |
33.93±0.88 |
31.78±0.58 |
31.38±0.51 |
30.43±0.70 |
|
4 |
91.32±1.30 |
45.62±1.10 |
44.35±1.14 |
44.86±0.97 |
39.99±0.81 |
38.84±0.59 |
38.04±0.82 |
|
5 |
92.68±2.92 |
55.34±1.57 |
53.41±1.73 |
54.22±1.91 |
52.25±1.89 |
51.20±0.88 |
50.10±1.19 |
|
6 |
92.88±3.12 |
69.07±2.20 |
68.21±2.01 |
67.39±2.33 |
61.61±2.11 |
61.26±1.52 |
60.01±2.28 |
|
7 |
93.00±2.87 |
78.69±2.55 |
77.36±1.99 |
77.97±2.21 |
71.72±1.99 |
70.67±2.19 |
69.52±2.33 |
|
8 |
93.12±1.98 |
89.27±3.01 |
87.69±2.57 |
88.40±2.49 |
77.13±2.37 |
76.31±2.26 |
76.03±2.40 |
|
12 |
93.24±3.42 |
88.91±3.71 |
88.25±3.41 |
88.76±2.52 |
86.29±2.82 |
86.19±2.33 |
84.98±2.51 |
|
24 |
93.28±3.29 |
89.32±3.28 |
89.01±3.25 |
88.96±2.56 |
87.49±3.01 |
86.74±2.32 |
86.14±2.53 |
The release study must be upto 12 h
We have changed.
Respectfully
Laura Vicas et al
Round 2
Reviewer 1 Report
In the revised manuscript, the pictures displayed are still unclear. Personally, the work in the article failed to provide enough new information.
Author Response
Esteemed reviewer,
Thank you for your comments.
Response to Reviewer 1 Comments
In the revised manuscript, the pictures displayed are still unclear. Personally, the work in the article failed to provide enough new information.
The observations were taken into account.
We changed the unclear pictures and the structure of manuscript.
Respectfully
Laura Vicas et al
Reviewer 3 Report
Major concerns of me have been settled well such as the tedious introduction and the confusing hydrogel formulae. There's still several suggestions that might improve the overall quality of this work.
- Photographs of hydrogels are usually presented instead of verbal description, such as "homogeneous, translucent appearance, yellowish-brown color" in section 2.2;
- Qualities of Figuresneed further improvement, the annotations could barely be read in the PDF file;
Author Response
Esteemed reviewer,
Thank you for your comments.
Major concerns of me have been settled well such as the tedious introduction and the confusing hydrogel formulae. There's still several suggestions that might improve the overall quality of this work.
- Photographs of hydrogels are usually presented instead of verbal description, such as "homogeneous, translucent appearance, yellowish-brown color" in section 2.2;
We added a figure
2. Qualities of Figures need further improvement, the annotations could barely be read in the PDF file;
We have made the suggested modifications.
Respectfully
Laura Vicas et al